# Nitrate Supplementation Combined with a Running Training Program Improved Time-Trial Performance in Recreationally Trained Runners

**DOI:** 10.3390/sports7050120

**Published:** 2019-05-21

**Authors:** Jeferson Santana, Diana Madureira, Elias de França, Fabricio Rossi, Bruno Rodrigues, André Fukushima, François Billaut, Fabio Lira, Erico Caperuto

**Affiliations:** 1Department of Physical Education, University São Judas Tadeu, São Paulo, SP 05503-001, Brazil; dianamsantos08@gmail.com (D.M.); defranca91@gmail.com (E.d.F.); ericocaperuto@gmail.com (E.C.); 2Immunometabolism of Skeletal Muscle and Exercise Research Group, Department of Physical Education, Federal University of Piauí (UFPI), Teresina, PI 64049-550, Brazil; fabriciorossi@ufpi.edu.br; 3Associate Graduate Program in Health Science, Federal University of Piauí (UFPI), Teresina, PI 64049-550, Brazil; 4School of Physical Education, University of Campinas (Unicamp), Campinas, SP 13.083-051, Brazil; prof.brunorodrigues@gmail.com; 5Department of Pathology, School of Veterinary Medicine and Animal Science, University of São Paulo (USP), São Paulo, SP 05508-010, Brazil; fukushima@usp.br; 6Department of Research and Extension, Igesp Health Sciences College (Fasig), São Paulo, SP 01301-000, Brazil; 7Département de kinésiologie, Université Laval, Québec, QC G3C 0S6, Canada; francois.billaut@kin.ulaval.ca; 8Exercise and Immunometabolism Research Group, Department of Physical Education, São Paulo State University (UNESP), Presidente Prudente, SP 19060-900, Brazil; fabioslira@gmail.com

**Keywords:** sport nutrition, endurance training, performance, nitrate

## Abstract

Our purpose was to verify the effects of inorganic nitrate combined to a short training program on 10-km running time-trial (TT) performance, maximum and average power on a Wingate test, and lactate concentration ([La^−^]) in recreational runners. Sixteen healthy participants were divided randomly into two groups: Nitrate (*n* = 8) and placebo (*n* = 8). The experimental group ingested 750 mg/day (~12 mmol) of nitrate plus 5 g of resistant starch, and the control group ingested 6 g of resistant starch, for 30 days. All variables were assessed at baseline and weekly over 30 days. Training took place 3x/week. The time on a 10-km TT decreased significantly (*p* < 0.001) in all timepoints compared to baseline in both groups, but only the nitrate group was faster in week 2 compared to 1. There was a significant group × time interaction (*p* < 0.001) with lower [La] in the nitrate group at week 2 (*p* = 0.032), week 3 (*p* = 0.002), and week 4 (*p* = 0.003). There was a significant group time interaction (*p* = 0.028) for Wingate average power and a main effect of time for maximum power (*p* < 0.001) and [La^−^] for the 60-s Wingate test. In conclusion, nitrate ingestion during a four-week running program improved 10-km TT performance and kept blood [La^−^] steady when compared to placebo in recreational runners.

## 1. Introduction

Nitrate is an inorganic anion present in the environment in various foods, especially vegetables such as celery, beetroot, lettuce and spinach [1,2]. After its consumption, nitrate circulates through the plasma, with an average half-life of 5 h. After it is absorbed in the blood, about 25% returns to the salivary glands, through an active transport, and concentrates in the saliva, with the rest being excreted by the kidneys. Nitrate concentrated in saliva is converted to nitrite by facultative commensal bacteria, which reside in crypts on the surface of the tongue. After that, this nitrite can be converted to nitric oxide in the stomach, due to its acidity becoming available to the organism [1,3,4,5].

The daily doses of 4.1 mmol to 16.8 mmol (approximately 250 mg to 1 g) of nitrate, consumed from 2 to 15 days, increase nitrite levels in the blood [1,6,7,8]. A review article showed that the typical averages used in studies range from 5 mmol to 9 mmol (approximately 300 mg to 550 mg) [1]. Nitrate is consumed usually between 1.5 h and 3 h before exercise, in a single dose up to five times per day [6,7,9,10,11].

Studies have shown that nitrate supplementation promotes vasodilatation, increases blood flow to the muscle, favoring the uptake of nutrients in the skeletal muscle and muscle contraction, attenuating the release of excess calcium, and subsequently reducing the ATP production cost [12,13]. Larsen et al. [14] showed that 0.1 mmol/kg/day of sodium nitrate supplementation in isolated skeletal muscle mitochondria promotes higher respiratory control than mitochondria from non-supplemented controls. Lansley et al. [15] showed that nitrate supplementation (6.2 mmol of nitrate in beetroot juice) for six days decreased 7% the oxygen amount required for constant rate moderate work and 15% in severe intensity running. Therefore, the low cost of oxygen in exercises with submaximal intensity, the greater mitochondrial efficiency, and physiological responses of fast twitch fibers (type II fibers), which can reduce NO_3_ to NO_2_, improving local perfusion, fatigue resistance, and muscle fiber contraction could improve performance of runners in sprint races [11,16,17].

Some investigations analyzed the performance effects of nitrate supplementation on runners. The meta-analysis from Hoon et al. [18] demonstrated that the nitrate supplementation had a minor benefit on time-trial (TT) performance and graded exercise tests in trained participants. Jones (2014) also showed that greater effects occurred when nitrate was ingested chronically and exercise was less than 30 min, such as short bouts or sprints. de Castro et al. [19] verified the effects of chronic nitrate supplementation on 10-km running performance in recreational runners and they administered 420 mL of beetroot juice for three days, and on the days of the assessments, the ingestion occurred 2 h before the test. The authors observed lower time to complete the first half of the test (5 km) compared to placebo, however, there were no statistically significant difference in the performance of the 10-km run. Thus, it seems that trained participants usually present minor responses to nitrate supplementation [20]. Therefore, there was no investigation of the ergogenicity of nitrate ingestion during a training program. One may hypothesize that the chronic ingestion of nitrate may enhance the responses to training, and, thereby, further improve performance.

Therefore, the purpose of the present study was to verify the effects of inorganic nitrate supplementation combined with a periodized running program on two different outcomes; the primary was 10-km running TT performance and the power developed during a Wingate test, and the second was the [La^−^] in the mentioned tests in recreationally trained runners.

## 2. Material and Methods

### 2.1. Experimental Approach to the Problem

This study was a randomized trial, double-blind design. Before any intervention, all participants signed the informed consent, and the study was approved by the Research Ethics Committee. Then, the participants were divided randomly into a nitrate or placebo group. Participants performed the same running training protocol during the study (described below). The participants completed a 60-s Wingate Test and a 10-km running TT at baseline and weekly during the training program. The Wingate test and the TT were conducted at the same time of the day and on the Friday and the Sunday of every week, respectively. This study has an experimental design similar to other studies already published by our group, including the study of Santana et al. [21].

### 2.2. Participants

Twenty healthy men were selected for this study. Participants with a minimum of 6 months of experience, with personal best times for 10 km between 55 and 65 min, and who trained 3 running sessions weekly were selected for the study. No supplements or ergogenics (different from the prescribed) were allowed during the study.

Participants that performed less than 75% of the training sessions, that did not use supplementation as indicated by the authors, that changed their usual diet, and did not present a medical approval form in the first week of the protocol were excluded.

Four participants were excluded after the end of the study because they missed more than 25% of the training sessions (exclusion criteria). No participants dropped out of the study, but we analyzed only the 16 participants remaining (Table 1), performing a per-protocol set analysis.

The University Ethical Committee approved the described experiments (protocol number CAAE: 38414814.3.0000.0089). All participants included in the study agreed to participate in the study and signed the consent form.

### 2.3. Procedure

#### 2.3.1. Experimental Protocol

Nitrate and placebo supplements were supplied for 30 days using a double-blinded method. All supplements were produced in a compound pharmacy (Orion Compound Pharmacy-São Paulo-Brazil) that gave us bottles with A or B marked on it and then we randomized the distribution considering the performance times of the participants.

The participants randomization was made by two members of our research lab (that were not part of this study) that offered both types of bottles, initially to the 4 slowest runners (those closest to 65 min to complete the 10-kmTT) and so on, to the 4 fastest runners of the group. Once the participant chose a bottle marked with a letter, he should only pick the same letter through the entire experimental protocol.

We understand that this way, we had distributed equally and blinded both supplements through the participants’ entire performance time interval and this distribution produced unbiased data.

The participants that received the placebo (*n* = 8) ingested 6 g of resistant starch in capsules, divided into three times a day, and the ones who received nitrate (*n* = 8) ingested 750 mg of inorganic nitrate and 5 g of resistant starch in capsules, divided into three times a day. The participants received capsules with supplements (nitrate or placebo) each week during the intervention, totaling 30 doses in the month [21].

There was no difference in the supplement consumption compliance. Both groups consumed more than 75% of the programmed supplementation in the month (nitrate 29.25 ± 0.9; placebo 29.63 ± 0.5 doses; *p* = 0.402).

#### 2.3.2. Wingate Test and Blood Lactate Concentration

Participants performed a five-minute pre-load with 80% of the maximum heart rate of each individual and, after that, they did a 60 s lead-in on a Wingate bike in order to generate the highest possible power in that period of time, with a load of 2% of the total body weight of the participants [22]. Participants were instructed to wear running clothes and the same running shoes in every test. [La^−^] was measured using a Roche portable lactate analyzer (Hoffmann–La Roche, Basileia, Switzerland). The analyses were carried out immediately after the 60 s Wingate test.

#### 2.3.3. Running Time Trial and Blood Lactate Concentration

The 10-km running TT was performed two days after the Wingate test so there was no interference between the tests. Just as the Wingate test, participants were instructed to wear running clothes and the same running shoes in every test.

Tests were executed at the same time, temperature, and climate conditions, according to the local weather forecast information, on an outside running track familiar to the participants. All tests started in the middle of August and finished in the end of September, taking around 45 days; this was the time of a reasonably steady climate condition.

The tests were performed on a running track familiar to the participants, with all external factors interfering, to be as similar as possible to a running competition.

A member of the study recorded time in the running test. Start and finish of the test were in the same place, with all the participants running together; they were not told to compete against each other but that happened, naturally. [La^−^] was measured as described, through the collection of a drop of blood from the fingertip on a reagent strip using a Roche portable Wingate test [21].

#### 2.3.4. Training Protocol

Participants received a periodized training program of 4 weeks (30 days), involving three running sessions weekly (Monday, Wednesday, and Friday), totaling 12 training sessions in the month. On the first training of each week, participants ran a moderate volume session (5 to 7 km). On the second day, the participants trained 4 to 6 sprints of 500 m at high speed with 2 min passive recovery between sprints. On the third day, the participants ran a long-distance (9 km to 12 km) session. Participants were supervised all day by the author of the study, and heart rate was monitored by the participants [21].

There was no difference in the training sessions compliance. Both groups did more than 75% of the training sessions through the month (nitrate 11.75 ± 0.5; placebo 11.63 ± 0.7; *p* = 0.598).

## 3. Statistical Analyses

A 2 × 5 (group × time) repeated measures analysis of variance with Bonferroni adjustment for multiple comparisons was used to compare placebo and nitrate group on the performance and [La]. Statistical significance was set at *p* < 0.05. For all measured variables, the estimated sphericity was verified according to Mauchly’s W test and the Greenhouse–Geisser correction used when necessary. The effect size was also calculated via partial eta-squared for main effect of time (ES). The data were analyzed using the Statistical Package for Social Sciences 17.0 (*SPSS* Inc., Chicago, IL, USA).

## 4. Results

Table 1 presents the mean and standard deviation values for age, body weight, height, fat-free body mass, and % fat at baseline in the placebo and nitrate groups. There were no statistically significant differences between groups at baseline for any variables investigated, reassuring the efficiency of the randomization protocol.

Figure 1 shows the time to complete the five 10-km running TTs and the power developed in the 60-s Wingate tests for placebo and nitrate groups.

We observed a main effect of time (F = 21.302, *p* < 0.001, ES = 0.68) and significant group × time interaction (F = 13.387, *p* < 0.001) for the time to complete the TT. Post-hoc analysis revealed that time decreased significantly in all timepoints compared to baseline, and, more importantly, that the nitrate group was faster (−3.2 min) in week 2 than week 1, whereas there was no difference in the placebo group. There were no significant differences in the baseline values comparing both groups.

For maximum power developed during the Wingate test, there was a main effect of time (F = 12.641, *p* < 0.001, ES = 0.47), but no interaction (F = 1.869, *p* = 0.129) or significant difference between group were observed (F = 3.028, *p* = 0.104). There was, however, a significant group × time interaction (F = 2.934, *p* = 0.028) for average power, but the post-hoc analysis did not identify significant difference between groups. There were no significant differences in the baseline values comparing both groups.

Figure 2 presents the differences in the lactate concentration in the 10-km running test and the 60-s Wingate test weekly between placebo and nitrate groups.

For lactate during 10-km running, there was a significant group × time interaction (F = 5.943, *p* < 0.001) with lower [La^−^] in the nitrate group (F = 4.942, *p* = 0.043) compared to placebo at week 2 (*p* = 0.032), week 3 (*p* = 0.002), and week 4 (*p* = 0.003). Post hoc analysis also showed higher [La^−^] in the placebo group after week 3 compared to week 1 (*p* = 0.006) and week 5 in relation to baseline (*p* = 0.044) and week 1 (*p* = 0.011).

During the Wingate test, there was a main effect of time (F =16.891, *p* < 0.001, ES = 0.55), but no group × time interaction (F = 2.368, *p* = 0.064).

There were no significant differences in the baseline values comparing both groups for these two variables.

## 5. Discussion

Several studies have examined the acute and short-term chronic impact of nitrate supplementation in different exercise modalities. To our best knowledge, this study is the first to analyze the effects of a four-week periodized training program supplemented with daily nitrate ingestion on a 10-km running TT and power developed during a 60-s maximal cycling test in recreational runners. The main finding of this study is that nitrate supplementation improved performance within 7 days and induced an additional improvement at 14 days in a 30-day protocol. This performance improvement was concomitant with a lowering in blood [La^−^] during exercise within 14 days. Nitrates did not, however, change the performance indices associated with more “anaerobic” performance.

Plasma nitrite levels are a highly sensitive marker of NO bioavailability. Several researchers showed that nitrate supplementation increases plasma nitrite levels at low doses (4.1 mmol) and high doses (16.8 mmol) [6], both in acute supplementation and in chronic supplementation [8,23] proving that nitrate supplementation increases nitric oxide concentration. Thus, a supplementation of 750 mg of nitrate (approximately 12 mmol), as used in this study, is likely to increase the concentration of nitric oxide. The current results of enhanced performance during an approximately 58 min long exercise add to the literature. In fact, a systematic review published by Dominguez et al. [24] showed that nitrate supplementation improves cardiorespiratory endurance in exercises shorter than 30 min. Another meta-analysis showed that nitrate supplementation, despite presenting smalls effects, had significantly greater effect size when compared to a placebo control in a time-to-exhaustion protocol, but in time-trial performance, the small effects of the nitrate supplementation were not significant [25]. Usually, studies with exercises over 30 min are performed with athletes. Previous studies have shown that men with VO_2máx_ > 60 mL·kg^−1^·min^−1^ and women with VO_2máx_ > 50 mL·kg^−1^·min^−1^ do not present significant enhancements with nitrate supplementation because these participants have a high level of fitness or have high levels of nitrate or nitrite in the blood, with a lower response with the use of this supplement [20,23,26]. Since our study was performed with recreational runners, the lower fitness may explain the positive effects reported here.

Possibly the greatest aspect of this study is the weekly analysis of the impact of nitrate supplementation over the course of a four-week periodized training program. Previous investigations have used acute (2–2.5 h prior to exercise) or short-term (1 to 15 days) chronic nitrate supplementation [27]. Investigations of longer supplementation periods with weekly performance analysis are scarce. Wylie et al. [28] used doses of 3 mmol and 6 mmol of nitrate over 30 days, with analyses at 7, 28, and 30 days, and showed that the largest dose can reduce submaximal exercise O_2_ cost. The decrease in the cost of O_2_ might be associated with greater running economy and consequently improved performance over long distances. Those results would help explain the data in our study [29]. In addition, our data show a performance improvement in the 7th day, results similar to those found by Wylie et al. [28], but also demonstrates an additional improvement in the 14th day. These data bridge a gap in the literature since the performance improvement results with chronic nitrate supplementation were reported to occur between 1 and 15 days or after 28 to 30 days of supplementation, with no result being registered between 15 and 28 days of supplementation. This helps refine current supplementation recommendation for moderate-training runners.

Previous studies have shown that running long distances increase blood [La^−^], decreasing performance, although lactate is not the cause of fatigue per se [30,31,32]. A study from our research group showed that high blood [La^−^] are associated with impaired performance in a 10-km running time trial, corroborating this information [21]. Nitrate supplementation did not change blood [La^−^] compared to placebo in well-trained 10-km runners as shown in a similar study [23]. However, highly trained participants or athletes compared to recreational runners presented lower [La^−^] with nitrate supplementation given in an acute manner (~12.5 mmol acute supplementation), which is different from our study, since it was made with recreational runners with nitrate given in a chronic way, which can explain these conflicting results. Our results then suggest that nitrate could have shifted the energy supply from an anaerobic source to an oxidative supply, as stated by Wylie et al. [28]. These results indicate that the benefits of nitrate supplementation on blood [La^−^] in long-distance running or high-intensity exercises are dependent on the supplementation period/protocol (acute or chronic) and of the level of physical fitness of the subject.

We also examined changes in Wingate test performance since runners have been shown to enhance their running economy and endurance performance with high-intensity training [33,34]. Our study did not show significant effects on maximal power on Wingate Test in the recreational runners. Previous studies show contradictory results. Dominguez et al. [35] presented performance improvements in maximum power on a 30-s Wingate test with acute nitrate supplementation (~5.6 mmol) during the first 15 s. Other studies show no performance improvements in maximum power 30-s Wingate tests with six days of nitrate supplementation (~800 mg per day) [36]. Future studies are needed to better understand the mechanisms involved with anaerobic power and nitrate supplementation.

A possible explanation of this result is that the participants in this study selected aerobic exercise as their activity of choice, with greater adaptation on slow-twitch fibers and smaller adaptation in fast-twitch fibers, probably not enough to be detected in the Wingate test. Furthermore, nitrate supplementation dosage, period of supplementation, subject fitness levels, and time of the Wingate test protocol (60 s vs. 30 s, which is usually used), might have interfered with our results.

The difference in fat free body mass (*p* = 0.071) between groups is another important element, although it was not significant.

### Limitations of Study

This study presents some limitations. We acknowledge that the number of participants was low and may not be fully representative of the runner population. However, the selection of participants was rigorous. The inclusion and exclusion criteria limited the number of participants, mainly due to the use of any supplements or ergogenic substances during the experimental protocol. Most runners did not want to stop using other supplements during the study.

Control of supplement intake and training was also a limitation of the study. Daily contact was made with everyone, but this contact was sometimes only by text messages, thus some training and supplement intake was not personally verified.

We did not perform the nitrite concentration analysis in the plasma. This analysis would give us further important information that would facilitate the understanding of our data.

## 6. Conclusions

We concluded that 30 days of nitrate supplementation added to a chronic aerobic training regimen improves 10-km running TT performance in recreational runners more than the same training alone, due to a steadier blood [La^−^]. Our data points out that nitrate supplementation improves 10-km running time trial within 7 days of supplementation and, again, within 14 days.

## Figures and Tables

**Figure 1 sports-07-00120-f001:**
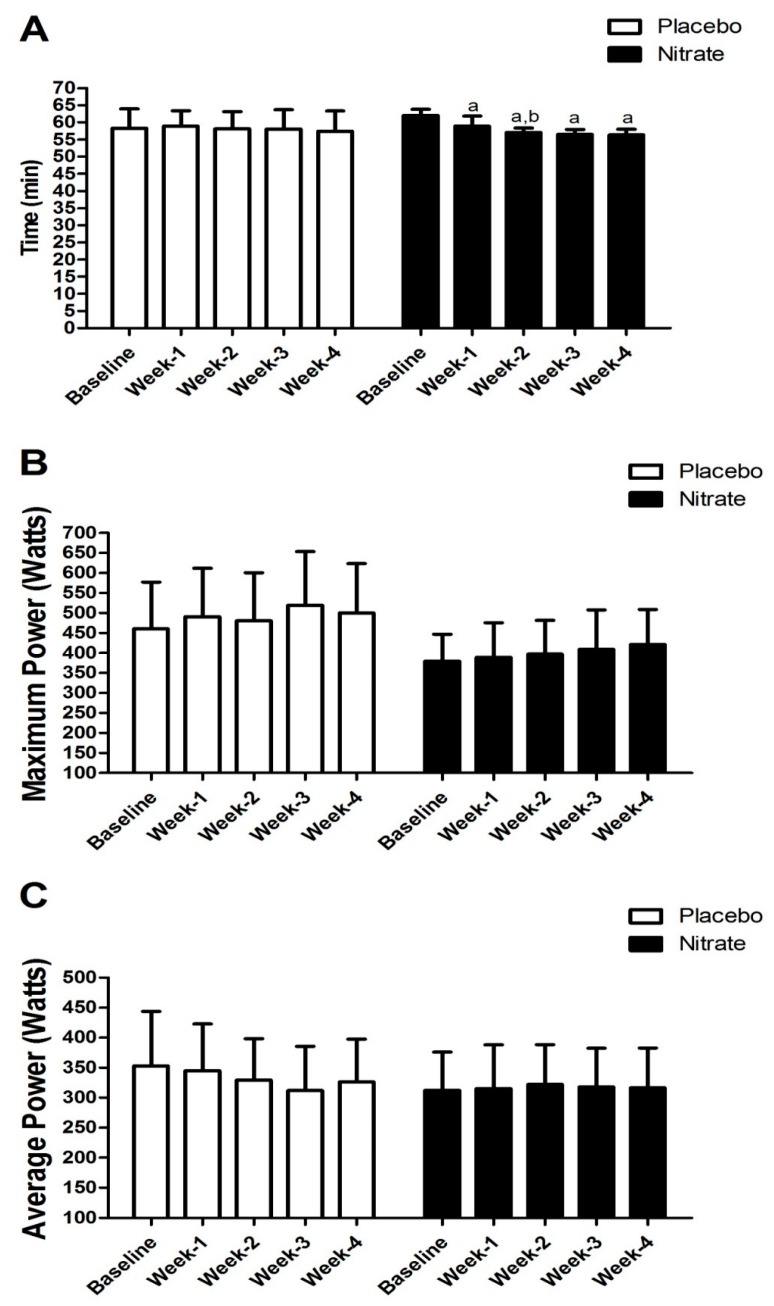
Comparison between placebo and nitrate groups for the time to complete the 10-km running time trials (panel 1-**A**), and maximum (panel 1-**B**) and average (panel 1-**C**) power developed during the Wingate test performed weekly during the training program. a is Bonferroni’s test result with *p* < 0.05 compared to baseline; b is Bonferroni’s test result with *p* < 0.05 compared to week 1.

**Figure 2 sports-07-00120-f002:**
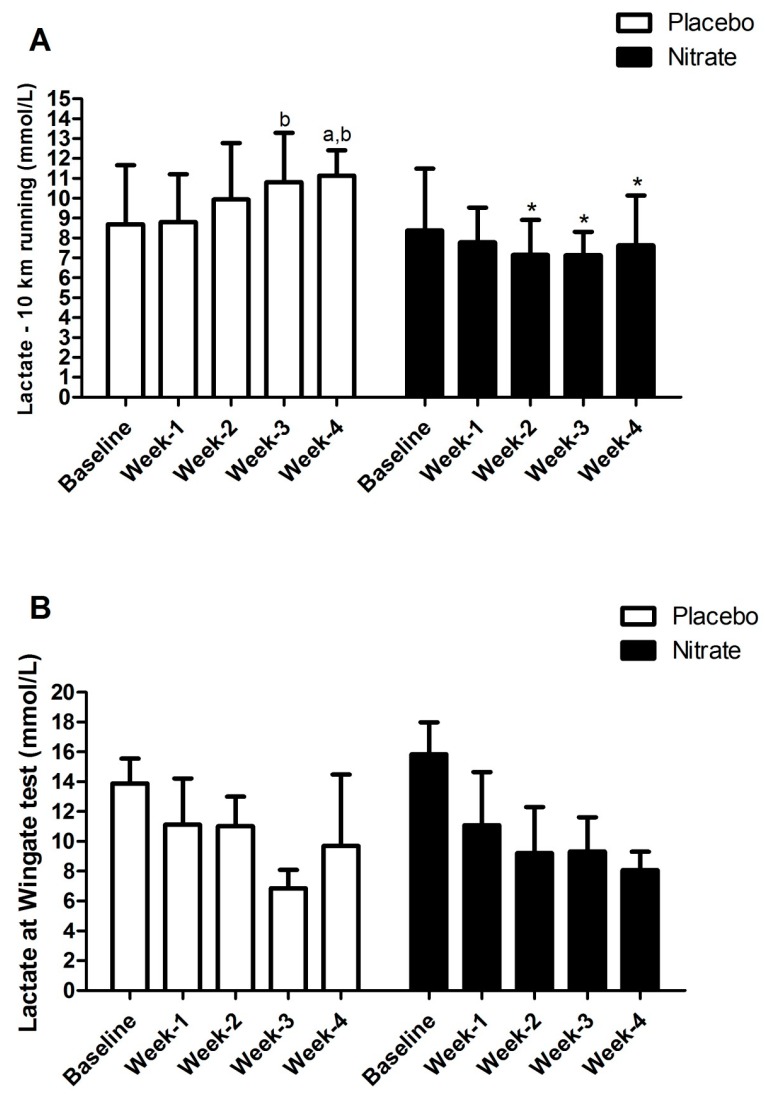
[La^−^] after the 10-km running time trial (TT) (panel 2-**A**) and the 60-s Wingate test (panel 2-**B**) for placebo and nitrate group. a is Bonferroni’s test result with *p* < 0.05 compared to baseline; b is Bonferroni’s test result with *p* < 0.05 compared to week-1; (*) Bonferroni’s test result with *p* < 0.05 compared to Placebo.

**Table 1 sports-07-00120-t001:** General characteristics of the participants at baseline.

Characteristics	Placebo (*n* = 8)	Nitrate (*n* = 8)	*p*
Age (years)	30.3 ± 4.5	30.0 ± 6.8	0.999
Height (m)	1.73 ± 0.1	1.66 ± 0.1	0.563
Weight (kg)	79.5 ± 11.2	68.8 ± 10.6	0.257
BMI (kg/m^2^)	25.65 ± 1.38	25.85 ± 1.84	0.793
Fat free body mass (kg)	63.7 ± 11.3	52.9 ± 8.5	0.071
% Fat	20.4 ± 4.5	23.3 ± 2.4	0.176

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
