# Peer review of "Nitrate Supplementation Combined with a Running Training Program Improved Time-Trial Performance in Recreationally Trained Runners"

_sports, 2019, doi:10.3390/sports7050120_

Reviewer 1 Report

This is a nice study which adds to the pool of literature on this topic. It is well written and clearly described. I have highlighted some specific comments which could be considered and may improve the paper in each section.

Abstract:

Line 21 subjects isn’t always appreciated by some as a term. Consider re-wording.

Training happened is a little clumsy. Consider re-wording

Introduction:

Mixed referencing style. Some numbered whilst others listed as authors. Correct throughout.

Line 58 NO2 and NO3. Subscript numbers.

Materials and Methods:

Good to see randomised and double blinded but it does look as if there may be some subtle differences between the two groups. Are they different for Height/Mass (not weight as described!)/BMI? P value may be useful here to clarify.

Clarify inclusion criteria. 55-65 minutes for 10K in 30-year-old males is a pretty low standard. Why was this chosen as ‘recreational’?

You state exclusion criteria but it would be nice to see some numbers with these statements. i.e. how many participants were initially recruited v’s how many dropped out or were excluded for noncompliance. This is useful information.

Why the choice of 60s Wingate and 2% body weight loading?

Clarify detail around the 10K time trial. Was this on a running track? Why not use treadmill/lab based for control? Did the participants complete this individually or did they race? You state running shoes but did they wear the same pair throughout all trials?

Training program. Why was this chosen? Was 3x per week additional to existing training these recreational runners were already doing? If not was it more or less than they were accustomed to? HR monitored during training?

Results:

Table 1: As per comment above P value to support statement around differences would be useful. It would also be nice to see any changes in body mass over the training period (if this was monitored).

If body mass was measured each week consider investigating Wingate data relative to BM as the lower values evident in the nitrate group is likely due to smaller stature.

Line 157: Do you mean -3.2 minutes rather than seconds? 3.2 secs on a 58-minute run can’t be a significant improvement (0.09%)

Discussion:

Same referencing issue as above.

It would be nice to see some discussion around the limitations of your study

Conclusion:

Word Probably is noncommittal and vague.

Last line ‘day 21 days’ needs rewording. 

Author Response

Abstract: 

Line 21 subjects isn’t always appreciated by some as a term. Consider re-wording.  

Ans. Revised in the manuscript to “participants” 

Training happened is a little clumsy. Consider re-wording 

Ans. Revised in the manuscript to took place. 

Introduction: 

Mixed referencing style. Some numbered whilst others listed as authors. Correct throughout. 

Ans. Revised in the manuscript 

Line 58 NO2 and NO3. Subscript numbers. 

Ans. Revised in the manuscript  

Materials and Methods: 

Good to see randomised and double blinded but it does look as if there may be some subtle differences between the two groups. Are they different for Height/Mass (not weight as described!)/BMI? P value may be useful here to clarify. 

Ans. Thank you for these comments. The term “weight” was corrected. BMI was not different between groups; we have now added the P value in Table 1. 

Clarify inclusion criteria. 55-65 minutes for 10K in 30-year-old males is a pretty low standard. Why was this chosen as ‘recreational’? 

Ans. We did not have access to well-trained runners for this study. The participants trained at least 3 times per week, they had long time of experience, but owing to their 10km time, we classified them as “recreational”.  

You state exclusion criteria but it would be nice to see some numbers with these statements. i.e. how many participants were initially recruited v’s how many dropped out or were excluded for noncompliance. This is useful information.  

Ans. Comment acknowledged. The information has been added to the manuscript. 

Why the choice of 60s Wingate and 2% body weight loading? 

Ans. We chose this Wingate protocol because we considered it more similar to the training and the test the participants did. 

Clarify detail around the 10K time trial. Was this on a running track? Why not use treadmill/lab based for control? Did the participants complete this individually or did they race? You state running shoes but did they wear the same pair throughout all trials? 

Ans. Participants were instructed to use light shorts, light t-shirt, and the same pair of running shoes in every test. The tests were performed on an outdoor running 

track, with all external factors interfering, to be as similar as possible to a regular race. Start and finish of the test were in the same place, with all the participants running together, although we did not tell them to race each other, it happened naturally. These details have been added to the revised manuscript. 

Training program. Why was this chosen? Was 3x per week additional to existing training these recreational runners were already doing? If not was it more or less than they were accustomed to? HR monitored during training? 

Ans. Participants were regular 10km runners. Their training was similar to the proposed training volume of the study, what we did was we equalized the volume of all participants. HR was monitored to evaluate intensity during the training sessions.  

Results: 

Table 1: As per comment above P value to support statement around differences would be useful. It would also be nice to see any changes in body mass over the training period (if this was monitored). 

Ans. P value was included in table 1. There were no differences in body mass during the training period (in the moments we analyzed). 

If body mass was measured each week consider investigating Wingate data relative to BM as the lower values evident in the nitrate group is likely due to smaller stature.  

Ans. Comment acknowledged. We included this information in the revised manuscript (please see lines 251-253) The difference in fat free body mass (p= 0.071) between groups is another important element, although it was not significant.   

Line 157: Do you mean -3.2 minutes rather than seconds? 3.2 secs on a 58-minute run can’t be a significant improvement (0.09%) 

Ans. Thank you for picking up this mistake. This is now revised in the manuscript.  

Discussion: 

Same referencing issue as above. 

Ans. Revised in the manuscript 

It would be nice to see some discussion around the limitations of your study 

Ans. Revised in the manuscript (lines 256 to 265) 

Conclusion: 

Word Probably is noncommittal and vague. 

Last line ‘day 21 days’ needs rewording. Ans. Revised in the manuscript

Reviewer 2 Report

Interesting paper, well described, valid methodolgy and reslts are sound.

Author Response

Thank you for your comments and review

Reviewer 3 Report

Santana et al. analyzed the effects of nitrate combined with running training program on exercise performance, such as time-trial in a randomized blinded study. The authors observed the significant improvement of time-trial performance by daily ingestion of nitrate. 

Basically, the authors should follow the CONSORT statement, which describes how to report human clinical trial.

Major points

1) The authors should describe the detail of this study. For example, what was the primary and secondary outcome of this study? When did this study start and finish?  Who was blinded? How did the authors divide and randomize the participants? I recommend the authors to read the CONSORT statement.

2) The authors should describe how many people were recruited for the screening of this study, and how many people were analyzed. Was there no dropout in this study?

3) The authors should describe the compliance of two groups. If the compliance was low, the authors could not discuss the effects of nitrate.

4) In line 143, the authors described that BMI was shown in Table. 1, but it was not described in Table. 1. Furthermore, the authors should describe whether statistically significant differences in time, maximum power and average power were not observed between two groups in the baseline. In addition, the authors should describe the ratio of male and female in two groups.

5) The authors should describe the limitation of this study. Although the authors observed statistically significant differences, the limited number of participants would be one of the limitations of this study.

Minor points

6) The authors used two types of citation, [1, 2] and (Lansley (2011)). This should be unified throughout the manuscript.

7) The height (cm) in Table. 1 might be (m).

8) In Figure. 2, which statics was used for between-group differences shown as "*"?

9) The authors should disclose and describe the funding(s) of this study.

Author Response

Basically, the authors should follow the CONSORT statement, which describes how to report human clinical trial. 

Ans.  Thank you for this comment, but we considered this study as a randomized trial, and not a clinical trial because it was not register as such. However, we understand that the study is described according to the CONSORT statement elements. 

Major points 

1) The authors should describe the detail of this study. For example, what was the primary and secondary outcome of this study? When did this study start and finish?  Who was blinded? How did the authors divide and randomize the participants? I recommend the authors to read the CONSORT statement. 

Ans. Again, all the information that the reviewer mentions is in the manuscript, not exactly with those terms but in a way that ensure that the methodology is sound and the data is well presented to be replicated or used in a meta-analysis. We understand the importance of registering the study as a clinical trial and will start doing it from now on. But since this was not registered, we chose not to use the CONSORT statement per se.  

2) The authors should describe how many people were recruited for the screening of this study, and how many people were analyzed. Was there no dropout in this study? 

Ans. Revised in the manuscript with more detail (please see lines 89 to 99). 

3) The authors should describe the compliance of two groups. If the compliance was low, the authors could not discuss the effects of nitrate. 

Ans. Revised in the manuscript with more detail, compliance was high (please see lines 89 to 99). 

4) In line 143, the authors described that BMI was shown in Table. 1, but it was not described in Table. 1. Furthermore, the authors should describe whether statistically significant differences in time, maximum power and average power were not observed between two groups in the baseline. In addition, the authors should describe the ratio of male and female in two groups. 

Ans. BMI was included in Table 1. We only highlighted the statistical differences with the usual statistic symbols, while the rest of the data showed no significant difference between groups. This study was carried out with males only. 

5) The authors should describe the limitation of this study. Although the authors observed statistically significant differences, the limited number of participants would be one of the limitations of this study. 

Ans. Comment acknowledged. Limitations were included as suggested. Lines 256 to 265. 

Minor points 

6) The authors used two types of citation, [1, 2] and (Lansley (2011)). This should be unified throughout the manuscript. 

Ans. Revised in the manuscript. 

7) The height (cm) in Table. 1 might be (m). 

Ans. Revised in the manuscript. 

8) In Figure. 2, which statics was used for between-group differences shown as "*"? 

Ans. Revised in the manuscript (line 173 and 174) 

9) The authors should disclose and describe the funding(s) of this study. 

Ans. Revised in the manuscript. This study had no funding except from the material provided by the University. We added it in line 280.

Round  2

Reviewer 3 Report

>Basically, the authors should follow the CONSORT statement, which describes how to report human clinical trial. 

>Ans.  Thank you for this comment, but we considered this study as a randomized trial, and not a clinical trial because it was not register as such. However, we understand that the study is described according to the CONSORT statement elements. 

The authors misunderstood my concern. My concern was not the registration, but the quality of data presentation.

>Major points 

>1) The authors should describe the detail of this study. For example, what was the primary and secondary outcome of this study? When did this study start and finish?  Who was blinded? How did the authors divide and randomize the participants? I recommend the authors to read the CONSORT statement. 

>Ans. Again, all the information that the reviewer mentions is in the manuscript, not exactly with those terms but in a way that ensure that the methodology is sound and the data is well presented to be replicated or used in a meta-analysis. We understand the importance of registering the study as a clinical trial and will start doing it from now on. But since this was not registered, we chose not to use the CONSORT statement per se.  

Again, the authors misunderstood my concern. These points were important for the reproducibility and the quality of data presentation. The authors should describe 1) what was the primary and secondary outcome of this study? In general, the most important outcome should be mentioned. 2) when did this study start and finish? This is also important because some seasonal effect might affect the results. 3) Who was blinded, participants, analyst, practitioner or author? 4) How did the authors divide and randomize the participants? If the author used an alternating method, this study should not be described as "randomized" study.

If the authors had already mentioned, please show exactly where the authors described.

>2) The authors should describe how many people were recruited for the screening of this study, and how many people were analyzed. Was there no dropout in this study? 

>Ans. Revised in the manuscript with more detail (please see lines 89 to 99). 

The authors misunderstood "screening" and "enrollment". My concern was the parent population of participants. If the authors recruited more than 20 participants, and performed screening, the authors should describe how many people were recruited for screening. In addition, if the authors excluded some participants from analysis, the authors should mention that the authors perform a per-protocol set analysis.

>3) The authors should describe the compliance of two groups. If the compliance was low, the authors could not discuss the effects of nitrate. 

>Ans. Revised in the manuscript with more detail, compliance was high (please see lines 89 to 99). 

The authors should describe the exact numerical value of compliance, and perform statistical analysis. If there was statistical significance in compliance between two groups, the authors could not make any conclusion in general.

>4) In line 143, the authors described that BMI was shown in Table. 1, but it was not described in Table. 1. Furthermore, the authors should describe whether statistically significant differences in time, maximum power and average power were not observed between two groups in the baseline. In addition, the authors should describe the ratio of male and female in two groups. 

>Ans. BMI was included in Table 1. We only highlighted the statistical differences with the usual statistic symbols, while the rest of the data showed no significant difference between groups. This study was carried out with males only. 

If the authors enroll only male participants, the authors should describe this point. 

Because time, maximum power and average power were the outcomes of this study, it was important to mention that these parameters in baseline were equivalent between two groups. This reviewer had a strong concern that a baseline score of maximum power was different between the two groups. If there was no statistical significance, the authors should describe these points.

Author Response

>Basically, the authors should follow the CONSORT statement, which describes how to report human clinical trial. 

>Ans.  Thank you for this comment, but we considered this study as a randomized trial, and not a clinical trial because it was not register as such. However, we understand that the study is described according to the CONSORT statement elements. 

The authors misunderstood my concern. My concern was not the registration, but the quality of data presentation.

Ans. We would like to thank you for your time. All your suggestion and concerns are only elements that will make the study sounder and the data more understandable. We will try to elucidate and follow all your questions and suggestions.

>Major points 

>1) The authors should describe the detail of this study. For example, what was the primary and secondary outcome of this study? When did this study start and finish?  Who was blinded? How did the authors divide and randomize the participants? I recommend the authors to read the CONSORT statement. 

>Ans. Again, all the information that the reviewer mentions is in the manuscript, not exactly with those terms but in a way that ensure that the methodology is sound and the data is well presented to be replicated or used in a meta-analysis. We understand the importance of registering the study as a clinical trial and will start doing it from now on. But since this was not registered, we chose not to use the CONSORT statement per se.  

Again, the authors misunderstood my concern. These points were important for the reproducibility and the quality of data presentation. The authors should describe 1) what was the primary and secondary outcome of this study? In general, the most important outcome should be mentioned. 2) when did this study start and finish? This is also important because some seasonal effect might affect the results. 3) Who was blinded, participants, analyst, practitioner or author? 4) How did the authors divide and randomize the participants? If the author used an alternating method, this study should not be described as "randomized" study.

If the authors had already mentioned, please show exactly where the authors described.

Ans.  Comment acknowledged.

1) the primary outcomes were the time to complete the 10km trial and the Wingate test. Those were our performance evaluation parameters. Lactate test was a secondary outcome. Those were mentioned in the purpose (lines 76 and 77). We re wrote it so the outcomes are highlighted.

2) Our study took around 45 days (August to September) to collect all data. Surely there was some minor climate changes, but nothing significant. We added it to lines 133 and 134.

3) We understand that the participants and the authors were blinded about the supplements. We used a double blinded method. All supplements (placebo and Nitrate) were produced in a compound pharmacy that gave us bottles with A or B marked on it and then we randomized the distribution considering the performance times of the participants.

The participants randomization was made by two members of our research lab (that were not part of the study) that offered both types of bottles initially to the 4 slowest runners (those close to 65min. to complete the 10-kmTT) and so on to the 4 fastest runners of the group. This was a way that we understood we would have distributed equally and blinded both types of supplements through the entire performance time interval and this distribution produced unbiased data. We understand that this is not the best type of randomization possible, but we assumed that it would be enough to provide unbiased data.

We changed the way it was described to add all this detail about the randomization protocol. Please read lines 102 to 113.

We also added a line that connects the randomization protocol with the “no difference” baseline values. Please read lines 168 to 170.

4) OK, comment acknowledged. 

>2) The authors should describe how many people were recruited for the screening of this study, and how many people were analyzed. Was there no dropout in this study? 

>Ans. Revised in the manuscript with more detail (please see lines 89 to 99). 

The authors misunderstood "screening" and "enrollment". My concern was the parent population of participants. If the authors recruited more than 20 participants, and performed screening, the authors should describe how many people were recruited for screening. In addition, if the authors excluded some participants from analysis, the authors should mention that the authors perform a per-protocol set analysis.

Ans.  Comment acknowledged. We initially talked to more than 20 participants but some of them were not in the time interval required, some of them were supposed to be part of the study but didn’t show up, we didn’t record the exact number of initially recruited participants. We started and finished the study with 20 participants but we excluded 4 of them from the results because they missed more than 25% of the training sessions. The per-protocol set analysis information as well as a revised version of the screening and enrollment were added in lines 93 to 98.

>3) The authors should describe the compliance of two groups. If the compliance was low, the authors could not discuss the effects of nitrate. 

>Ans. Revised in the manuscript with more detail, compliance was high (please see lines 89 to 99). 

The authors should describe the exact numerical value of compliance, and perform statistical analysis. If there was statistical significance in compliance between two groups, the authors could not make any conclusion in general.

Ans.  Comment acknowledged. Revised in manuscript with more detail (please see lines 120 to 122 and 155 to 156).

>4) In line 143, the authors described that BMI was shown in Table. 1, but it was not described in Table. 1. Furthermore, the authors should describe whether statistically significant differences in time, maximum power and average power were not observed between two groups in the baseline. In addition, the authors should describe the ratio of male and female in two groups. 

>Ans. BMI was included in Table 1. We only highlighted the statistical differences with the usual statistic symbols, while the rest of the data showed no significant difference between groups. This study was carried out with males only. 

If the authors enroll only male participants, the authors should describe this point. 

Ans.  Comment acknowledged. It is described in line 89.

Because time, maximum power and average power were the outcomes of this study, it was important to mention that these parameters in baseline were equivalent between two groups. This reviewer had a strong concern that a baseline score of maximum power was different between the two groups. If there was no statistical significance, the authors should describe these points.

Ans.  Comment acknowledged. We understand the importance of this information. It was added on lines 175, 180, 196 and 197.

Round  3

Reviewer 3 Report

The authors fully answered my questions.